# Prognostic Assessment of the Performance Parameters for the Industrial Diesel Engines Operated with Microalgae Oil

**Sergejus Lebedevas** [1] **and Laurencas Raslavičius** [1,2,*]

1   Marine Engineering Department at Faculty of Marine Technology and Natural Sciences, Klaipeda University, LT-91225 Klaipėda, Lithuania; sergejus.lebedevas@ku.lt
2   Department of Transport Engineering, Faculty of Mechanical Engineering and Design, Kaunas University of Technology, LT-51424 Kaunas, Lithuania
*   Correspondence: laurencas.raslavicius@ktu.lt

**Abstract:** A study conducted on the high-speed diesel engine (bore/stroke: 79.5/95.5 mm; 66 kW) running with microalgae oil (MAO100) and diesel fuel (D100) showed that, based on Wibe parameters ($m$ and $\varphi_z$), the difference in numerical values of combustion characteristics was ~10% and, in turn, resulted in close energy efficiency indicators ($\eta_i$) for both fuels and the possibility to enhance the $NO_x$-smoke opacity trade-off. A comparative analysis by mathematical modeling of energy and traction characteristics for the universal multi-purpose diesel engine CAT 3512B HB-SC (1200 kW, 1800 min$^{-1}$) confirmed the earlier assumption: at the regimes of external speed characteristics, the difference in $P_{me}$ and $\eta_i$ for MAO100 and D100 did not exceeded 0.7–2.0% and 2–4%, respectively. With the refinement and development of the interim concept, the model led to the prognostic evaluation of the suitability of MAO100 as fuel for the FPT Industrial Cursor 13 engine (353 kW, 6-cylinders, common-rail) family. For the selected value of the indicated efficiency $\eta_i = 0.48$–$0.49$, two different combinations of $\varphi_z$ and $m$ parameters ($\varphi_z = 60$–$70$ degCA, $m = 0.5$ and $\varphi_z = 60$ degCA, $m = 1$) may be practically realized to achieve the desirable level of maximum combustion pressure $P_{max} = 130$–$150$ bar (at $\alpha$~2.0). When switching from diesel to MAO100, it is expected that the $\eta_i$ will drop by 2–3%, however, an existing reserve in $P_{max}$ that comprises 5–7% will open up room for further optimization of energy efficiency and emission indicators.

**Keywords:** transport decarbonization; hard-to-decarbonized modes; diesel engine family; heavy duty engines; microalgae oil; prognostic assessment; sustainability

## 1. Introduction

Today, the world is challenged with the twin crises of fossil fuel reduction and environmental degradation. Unselective extraction and excessive consumption of fossil fuels have led to a decrease in underground-based carbon capitals. The hunt for alternative energy, which assures a positive correlation with sustainable growth, energy conservation and management, efficiency, and ecological protection, has become extremely marked over the last two decades.

The entire transport sector, including industries providing transportation, agree on the need to decarbonize traffic before 2050–2060 for the most developed and 2060–2080 for less developed economies [1,2]. Usually, this roadmap is primarily associated with the wider deployment of electric transport. It is likely that different energy vectors (CNG, LPG, synfuels, vegetable oils and biodiesel, GTL, H₂, electricity—see Table 1) will play a role in transport decarbonization [3]. If properly allocated to hard-to-decarbonized modes of transport, synfuels and sustainable biofuels, coupled with the direct electricity consumption through either electrified railways or battery electric vehicles, will all be important in the process of reducing 'carbon intensity' in transport [3]. The European Union member countries alone consume approximately a fourth of the petroleum exploited globally per year. Global consumption of petroleum products has been growing as a

result of the rapid development of Asian economies (China, India) as well. EU authorities have recently started referring to new pollution and climate change control measures more frequently. There is a unanimous consensus within the community on securing long-term clean energy supplies for Europe, in addition to the reduction of greenhouse gas (GHG) emissions from the transport and energy sectors. Lower environmental pollution and higher economic efficiency are probably the biggest advantages of fuel alternatives to gasoline and diesel. However, several studies [4,5] provide readers with the counter argument, that a massive replacement of combustion engine-powered vehicles by battery electric vehicles alone cannot deliver greenhouse gas reductions consistent with climate stabilization and, in the future, may lead to the depletion of key mineral deposits, such as magnesium and lithium. The producers of the high-power industrial diesel engines of low- and average-speeds see fuel flexibility and robustness (VLSFO/MGO, 20% $H_2$ in NG, biofuels, MeOH, NH3, $H_2$ as future alternatives) as the key advantage and offer a wide range of sector-specific scenarios, outlining the potential benefits of a particular fuel choice.

**Table 1.** Alternative fuels currently being heavily studied for transport applications.

| Types of Fuel | Description |
|---|---|
| Oils and biodiesels (including microalgae oil and biodiesel), BTL (biomass-to-liquid), and alcohol fuel [6–10] | Because they are produced in plants that chemically 'fix' or capture carbon dioxide, these types of fuel are characterized for their low environmental pollution. Nonetheless, the production of such fuels requires large amounts of energy if compared with gasoline or diesel fuel. |
| Gas-to-liquid (GTL) [11–13] | Used as a substitute for diesel fuel, as GTL leads to a significant reduction in air pollution from internal combustion engines. |
| Synthetic fuels (or synfuels) [14,15] | Production of synthetic diesel fuels obtained from biomass, household waste, and/or natural gas has begun approx. 20 years ago. Synfuels are not considered as alternative fuels since they do not require any modifications in the fueling infrastructure or engine design. However, synfuels expand the raw materials base as well as enhance biodiversity and restore the natural ecological balance due to their easy quality assurance during the production process. |
| Compressed natural gas (CNG) and liquefied natural gas (LNG) [16] | CNG and LNG both are highly functional and efficient type of fuel gas. Theoretically, natural gas resources are vast, if not taking into account the global geopolitics. If we take into consideration the geopolitical situation in the world, specialists believe they may be depleted by 2060. Hence, natural gas is playing a large role in near-future energy prediction. The advantage of CNG/LNG compared with other types of alternative fuel is lower $CO_2$ emission and a higher heating value (48.7 MJ/kg) in comparison to diesel fuel (42.6 MJ/kg). Currently, nearly all European automobile manufacturers offer natural gas-powered vehicles to the market. Most of them can run both on gasoline and on natural gas, however, bi-fuel engines lead to higher environmental pollution if compared with those running only on natural gas. |
| Liquefied petroleum gas (LPG) [17] | LPG (mainly propane and butane) is prepared by refining raw natural gas or crude oil and is a co-product of the refining process. This type of fuel is highly explosive. Moreover, LPG characteristics are different from those of the diesel fuel, which means that engines designed for both types of fuel are inefficient. |
| Hydrogen ($H_2$) [18] | Hydrogen in gaseous or liquid form may be used in conventional internal combustion engines. This type of fuel carries three times more energy than gasoline; however, density of the former is significantly lower even when compressed. Moreover, a significant amount of electrical energy is required for $H_2$ generation. |

There is a widespread consensus that diesel engine technology has not reached its full maturity and potential yet, in terms of efficiency or lower carbon impact [1]. This is an advanced technology through which synthetic, zero-emission fuels can be produced using only renewable energy and $CO_2$ [1]. Fuels such as $H_2$, GTL, CNG, 3rd generation microalgae fuels, and synthetic hydrocarbons that are made using energy from renewables or other low-carbon energy sources could play a role in multiple hard-to-decarbonize sub-sectors of global transportation [19]. There is also another key, future-oriented reason that

should motivate legislators and OEMs to keep diesel engines in the game: power-to-fuel or power-to-x [1,20]. Synthetic fuels made from carbon dioxide captured from the air or 3rd generation microalgae fuels made from $CO_2$ captured from industrial power plants can be successfully used as transportation fuels in conventional engines.

Gaps in the literature, which we are trying to fill. The goal of the overall transport sector is to largely decarbonize and move from 7.7 metric gigatons of emissions per year to 3-2 metric gigatons by mid-century (2050), while ensuring climate resilience. Based on IEA data, predicted global demand for fuel and energy by the transport sector will increase by 140%, 75%, and 70% in aviation, freight transport, and passenger cars, respectively, between 2000 and 2050 [20]. As for the EU transport sector, the agreement was obtained in 2018 that a 14% RES target by 2030, including the gradual phase out of crop-based biofuels from 7% in 2020 to 3.8% in 2030 and a 3.5% share of advanced biofuels of 2nd and 3rd generation. As described previously, all kinds of alternative fuels being heavily studied for transport applications today will all be essential in the process of reducing 'carbon intensity' in transport. The main advantage of the 3rd generation microalgae oil-powered heavy-duty engine over other alternative fuels is that such a vehicle can be relatively called '$CO_2$ neutral'—a feature which is characteristic to the very limited variety of fuels of the future (synfuels, power-to-x, etc.) [8]. Notwithstanding the large amount of research studies conducted so far, the majority of the dedicated works are limited to an assessment of diesel engines of a particular modification and presentation of the insights and recommendations obtained for the engine-specific scenario [6–8]. This situation usually leads to the disparity between the total quantity of recorded knowledge and the limited capacity of researchers, government bodies, and legislators to assimilate it as well as take action [6–8]. This was the main reason behind the prognostic assessment of the industrial diesel engine family for energy efficiency and $CO_2$ levels and to take a broader look at pure microalgae oil, a potentially carbon-free resource, as a candidate for future transportation energy mix. This study is a part of larger scale investigation conducted by co-authors in the field of transport decarbonization.

## 2. Materials and Methods

A simplified scheme, known as an 'experiment planning chart', for this investigation is outlined in Figure 1. Below is a description of the basic steps in this research process. A reliable set of CI engine parameters that could be transferable to the entire family of industrial engines for a possible switch to pure microalgae oil was obtained to examine factors that may contribute to the acceptance of or resistance to this carbon-neutral type of fuel.

### 2.1. Fuels and Test Engines

The methodological part discusses first the 'pilot' study as the first step of the entire research trial, and then the 'main study' [21]. The purpose of the pilot study was to extract a number of parameters (factors) from an 1Z engine map of microalgae oil use coupled with the assessment of smoke opacity and $CO_2$ levels obtained at various loads and an in depth analysis of the thermodynamic engine cycle simulation. Pure microalgae oil (MAO100), as a less investigated type of fuel, was subjected to CAT 3512B HB-SC and FPT Industrial Cursor 13 engines performance modelling. Conventional diesel fuel (D100) not containing a required 5% biodiesel additive was used as a reference fuel to compare and to contrast the engine indicators obtained for both fuels (Table 2).

Obtaining complex input parameters set (primary and derrivative) from 1Z engine performance on microalgae oil + AVL boost modelling of in-cylinder processes from a study published by one of co-authors in 2019 [21].

Model calibration and evaluation of its ability to generate a reliable set of outputs from a supplied set of obtained inputs.

The use of a calibrated model for the assessment of parameters for CAT 3512B HB-SC industrial engine running with microalgae oil. Energy efficiency, process parameters and $CO_2$ emission levels were simulated and compared with the experimental data (from a study published by one of the co-authors in 2021 [23]) for the same engine that was exploited on diesel fuel (as an interim concept).

With refinement and development of the interim concept, the model led to prognostic evaluation of suitability of microalgae oil as fuel for FPT Industrial Cursor 13 engine family

**Figure 1.** Experiment planning chart.

**Table 2.** Properties of test fuels.

| Parameter | Microalgae Oil | Diesel Fuel |
|---|---|---|
| Composition | 77.81 C, 11.71 H, 10.48 O | 86.20 C, 13.80 H |
| Density (at 15 °C), kg/m$^3$ | 915.8 | 834.5 |
| Viscosity (at 40 °C), mm$^2$ | 5.287 | 2.286 |
| LHV, MJ/kg | 36.85 | 42.80 |
| HHV, MJ/kg | 39.62 | 44.80 |
| Cetane number | 53.2 | 52.4 |

During the 'pilot' study, the sensitivity of microalgae oil to commercial diesel fuel, using the primary and derivative parameters of the bench test engine (type-1Z) that were published by one of the co-authors in 2019 [21], were assessed to build a data matrix, whose rows represent different repetitions of an experiment, and whose columns represent different kinds of data taken for each repetition. A more detailed description of the specific test procedures, methodologies applied, technical characteristics of the measurement equipment, and uncertainties of the measured parameters is presented in Ref. [21]. The derivative parameters necessary to build a calculation matrix are as follows: $P_e$, $p_{me}$, *BSFC*, net efficiency, mass air flow rate, exhaust gas temperature, air excess ratio, etc. They were established at three different loads and with the advancement or retardation of the start of fuel injection timing every 2 degrees of the crankshaft rotation angle from $-16°$ to $+4°$ relative to TDC. To obtain a full dataset for the mathematical modelling of thermochemical processes, the following derivative parameters were also assessed: in-cylinder pressure, pressure rise rate, in-cylinder temperature, temperature rise rate, and heat release rate. Technical specifications of the engines subjected to the 'pilot' and 'main' studies are presented in Table 3, while Figure 2 depicts the schemes of the discussed above type of engine.

**Table 3.** Specification of the engines.

| Parameter | Type or Value | | |
|---|---|---|---|
| Model | 1Z | CAT 3512B HD-SC | FPT Industrial Cursor 13 |
| Total displacement | 1.896 dm$^3$ | 51.800 dm$^3$ | 5.900 dm$^3$ |
| Compression ratio | 19.5:1 | 14.0:1 | 16.5:1 |
| Aspiration | Turbo | Turbo | Turbo |
| No. of cylinders | 4 | 12 | 6 |
| Type of injection | Direct | Direct | Common rail direct fuel injection |
| Bore/stroke | 79.5 mm/95.5 mm | 170 mm/190 mm | 135 mm/150 mm |
| Rating | 66 kW | 1200 kW | 353 kW |
| Coolant | Water | Water | Water |

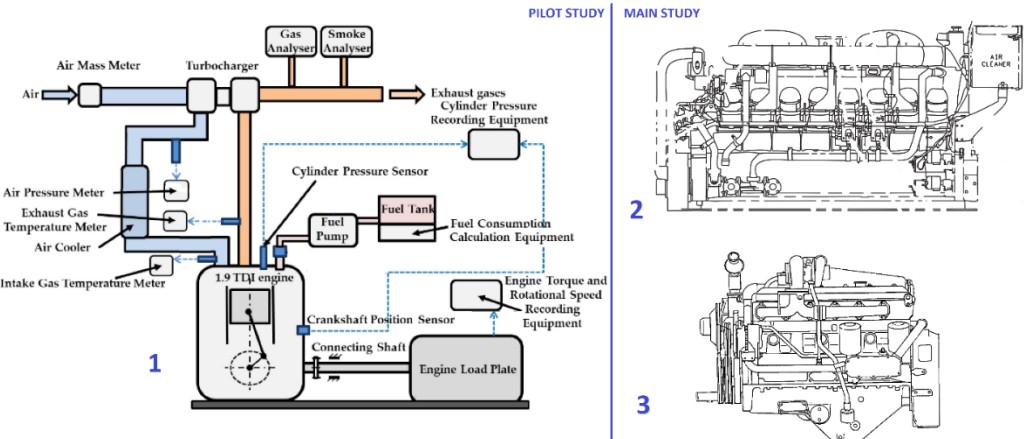

**Figure 2.** Schematic view of the pilot study equipment (**1**) and industrial engines being subjected to mathematical modelling: CAT 3512B HD-SC (**2**) and FPT Industrial Cursor 13 (**3**).

### 2.2. IMPULS Modelling

Mathematical model (MM) calibration by standard techniques required a primary (historical) database (an outcome of a pilot study) for key variables and the primary techniques of statistical process control. MM calibration was mainly carried out manually using many adjustments of the engine parameters obtained during its operation on microalgae oil to improve the model (both natural state and history matching).

A set of inputs used for adjusting the parameters of the 'Wiebe' [22] combustion function included in 1-D thermodynamic engine cycle simulation tool IMPULS, developed at the Central Diesel Research Institute (CNIDI) [23] was analyzed. The structure of this software is constantly being improved with the supplementation of sub-models intended to assess the formation and combustion of the fuel-air mixture, the dynamics of fuel injection, evaporation, flame spread, and the possibility to evaluate fuels with different chemical composition, etc. Most of the phenomenological sub-models implemented in this program are similar in their nature with the other widely used and adopted software—AVL BOOST [24]. The first law of thermodynamics for closed systems in the form of differential equation is formulated for the CI engine as follows:

$$\frac{dU}{d\tau} = \frac{dQ_{re}}{d\tau} - \frac{dQ_e}{d\tau} - p \cdot \frac{dV}{d\tau} + h_s \cdot \frac{dm_s}{d\tau} - h_{ex} \cdot \frac{dm_{ex}}{d\tau}, \ [\text{kJ/s}] \tag{1}$$

Therefore, $dU$ considers the change in internal energy in the system (J); $dQ_{re}$ quantifies the combustion heat released (J); $dQ_e$ calculates the energy exchange (wall heat transfer from the cylinder gas) (J); $pdV$ is the volumetric work ($p$—pressure (Pa); $V$—volume (m$^3$)); $h \cdot dm$ sum up all enthalpy flows (index $ex$ over the exhaust valves, $s$ over intake valves); $m_s$ is the supply (intake) air mass (kg); $m_{ex}$ is the mass of exhaust gas (kg); and $\tau$ is the time (s).

As described in Ref. [22], depending on the flow direction, a negative sign indicates that the enthalpy leaves the system, and a positive sign indicates an entering of enthalpy flow. The mass balance equation (Equation (2)) and state equation (Equation (3)) are as follows [25]:

$$\frac{dm}{d\tau} = \frac{dm_s}{d\tau} + \frac{dm_{inj}}{d\tau} - \frac{dm_{ex}}{d\tau}, \ [\text{kg/s}] \tag{2}$$

$$\frac{dp}{d\tau} = \frac{m \cdot R}{V} \cdot \frac{dT}{d\tau} + \frac{m \cdot T}{V} \cdot \frac{dR}{d\tau} + \frac{R \cdot T}{V} \cdot \frac{dm}{d\tau} - \frac{p}{V} \cdot \frac{dV}{d\tau}, \ [\text{Pa/s}] \tag{3}$$

where: $m_{inj}$ is the mass of injected fuel (kg); $R$ is the gas constant (J/kg·K); and $T$ is the temperature (K).

The rate of heat release according to the 'Wiebe' model [22,26] was determined by the following equation:

$$\frac{dx}{d\left(\frac{\tau}{\tau_z}\right)} = C(m+1)\left(\frac{\tau}{\tau_z}\right)^m \cdot e^{-C\left(\frac{\tau}{\tau_z}\right)^{m+1}}; \ dx = \frac{dQ}{Q} \tag{4}$$

Thereby, $Q$ describes the total heat input; $\tau$ is the angle between initial and current time; $C$ is a function parameter that is equal to 6.9 for the case of complete combustion; and $\tau_z$ is the relative time of combustion.

The fraction of heat released since the start of combustion can be assessed by using Equation (5) [26]:

$$x = 1 - e^{-C\left(\frac{\tau}{\tau_z}\right)^{m+1}} \tag{5}$$

The heat release was determined by the 'Wiebe' model, and parameters were then converted to partial load modes by using the empirical correlations by Woschni [22,27] in the form of a simplified relationship between the heat release parameters and operational parameters of an engine (fuel/air ratio, injection timing, crankshaft rotational speed, charge in air pressure, charge in air temperature, etc.). A simplified 'Wiebe' heat release diagram is characterized by two parameters: $m$-form factor and $\varphi_z$-conditional duration of combustion [23,27].

This set of equations is solved by using the selected method, which is programmed in the computer code together with the boundary conditions [26,28]. When the iterative process was finished, the validation of the measured and predicted results from the IMPULS model is performed. Experimental results in the form of datasets of a 1Z engine running with (i) microalgae oil and (ii) diesel fuel that were obtained during the pilot study were compared and contrasted with the ones derived by IMPULS software. The following parameters were compared at $p_{me}$ = 12.6, 18.9, and 25.1 bar: air pressure after compression ($p_k$), air temperature after compression ($T_K$), pressure of compression in the cylinder ($p_c$), maximum cycle pressure (combustion pressure) ($p_{max}$), exhaust gas temperature ($T_g$), hourly air supply ($G_{air}$), excess air coefficient ($\alpha$), in-cylinder pressure increase rate ($\lambda$), indicated thermal efficiency ($\eta_i$), and effective efficiency ($\eta_e$). A full description of each interim step of MM calibration to assess the 1Z engine running on pure diesel fuel is described in detail in Ref. [23]. Later on, MM suitability for accurate assessment of the complete cycle of combustion for a 1Z engine powered by MAO100 was investigated assuming that $\varphi_{inj}$ = 2 CAD BTDC, $n$ = 2000 min$^{-1}$, *BMEP* = 4, 6 and 8 bar. After checking the overlap between the experimental and modeling outcomes which show good agreement between the results, we made a prerequisite for the next step in this study—to transfer and to adopt the successfully established heat release characteristics for the 1Z engine of a passenger car to a heavy-duty industrial diesel engine equipped with a similar fuel injection system. Hence, the CAT 3512B HB-SC engine was assessed for six different speeds ($n$ = 1000, 1200, 1300, 1400, 1500, 1600, and 1800 min$^{-1}$). Variation in 10 parameters was analyzed for high load mode: brake effective mean pressure, indicated thermal efficiency, effective efficiency, brake-specific fuel consumption, exhaust gas temperature, excess air

coefficient, maximum cycle pressure (combustion pressure), pressure after compression, mechanical efficiency, and the change in $CO_2$ emissions.

The potential of this work extends beyond the creation of engine maps to allow investigations into the transferability of heat release characteristics from a passenger car diesel engine to the industrial one; this study offers a guideline as a modeling outcome for prognostic assessment of the engine parameters obtained for the entire family of engines [29,30]. Finally, modeling of the combustion process for the CI engine (see Table 3), representing the entire FPT Industrial Cursor 13 engine family has been performed to evaluate the compatibility of microalgae oil with a wide range of industrial CI engines. This part of the research was dedicated to further interpretation of engine operational parameters (combustion pressure, indicated efficiency, hourly fuel consumption, maximum combustion pressure, exhaust gas temperature, and $CO_2$ emissions) in order to describe the character of heat release according to various combinations of the form factor ($m$) and combustion duration ($\varphi_z$). A relative change in the aforementioned variables when switching an engine from diesel fuel to microalgae oil was established, as well as the zones of rational combination of $m$ and $\varphi_z$ for each specific parameter were identified. The fitness of this model for the entire family of industrial engines was demonstrated by the successful adoption of data sets obtained during the pilot study and their successful transfer creating candidate engine maps and identifying the zones of rational combination of $m$ and $\varphi_z$ parameters. On the basis of the tests and modelling conducted, the IMPULS model, due to the described above peculiarities of the algorithm realization, seems a potentially useful tool for compatibility analysis of various size diesel engines to run on microalgae oil with the described above physico-chemical properties (see Table 2).

## 3. Results

### 3.1. Selection of Strategies to Improve $NO_x$-Smoke Opacity Trade-Off

The findings of a pilot study showed that, although the compression ignition engines are designed for optimum operation with fossil fuel, further advancements in lowering exhaust gas emissions are possible through the use of $CO_2$-neutral fuel with favorable fuel characteristics and the proper adjustment of high reaction fuel injection time ($\varphi_{inj}$). Figure 3 shows the direction of possible $NO_x$-smoke opacity improvement for a 1Z engine. The abscissa denotes the specific emission of nitrogen compounds and the ordinate denotes the specific smoke opacity levels. The $NO_x$-smoke opacity trade-offs were assessed for the engine running at average load conditions, with the advancement or retardation of the start of fuel injection timing every 2 degrees of the crankshaft rotation angle (CA) from $-2$ to $+16$ degCA relative to the top dead center (TDC).

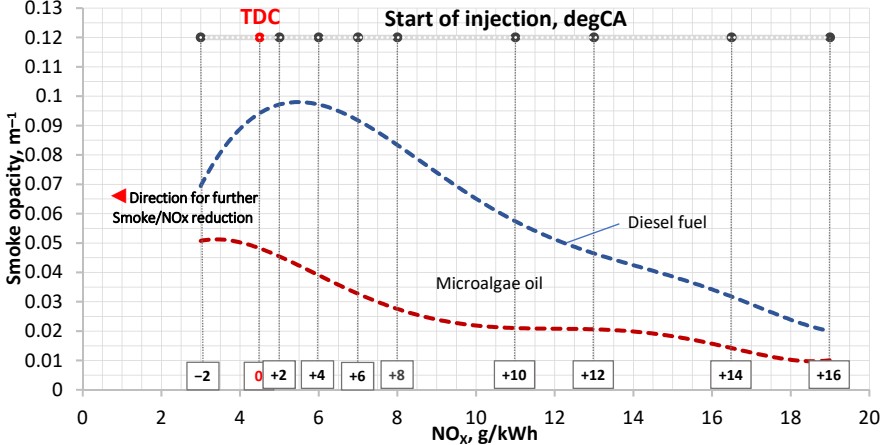

**Figure 3.** $NO_x$-smoke opacity trade-off for the different injection settings of a 1Z engine.

As shown in Figure 3, the original high values of smoke opacity in the case of diesel fuel operation are reduced by two-fold by replacing it with microalgae oil. As microalgae

oil produces lower smoke opacity levels compared with D100 at the same engine control setting, there is a big potential for nitrogen oxide emission reduction through a proper engine readjustment aiming to significantly improve the $NO_x$-smoke opacity trade-off. For the examined 1Z engine, the optimum approach was to retard the main injection timing (by 2 degCA) [31]. In that case, in addition to the reduction in smoke opacity levels predetermined by the simple replacement of diesel fuel with microalgae oil, further emission reductions were achieved leading to 41.2% (4.3 g/kWh vs. 2.5 g/kWh) lower engine-out $NO_x$ emissions for both unary fuels. Moreover, the interval of $-2 \ldots 0$ degCA can be described as the best setting of an engine for smoke and $NO_x$ stabilization and reduction: the first part of the interval related to 0 degCA demonstrated the same (and the second lowest) levels of $NO_x$ emissions for both tested fuels, while the second one showed a 22.2% and 2.0% decrease in smoke opacity levels for diesel fuel and MAO100, respectively. It is worth noting that, initially, the difference between smoke emission levels at 0 degCA for D100 and MAO100 was 49%, while the retardation of the SOI by two degrees relative to TDC reduced this difference to 29% (see Figure 3).

The findings presented in Figure 3 coincide well with the published works [32–34], which reported that one of the most significant CI engine parameters affecting the $NO_x$ emissions is retarding injection timing. As far as eight different advanced injection timings at 2000 $min^{-1}$ are concerned (from +2 to +16 degCA), the earlier was the injection timing, the lower were the $NO_x$ emissions due to lower in-cylinder temperature and heat release rate peak of the injection. The difference between the $NO_x$ levels obtained at +2 degCA and +16 degCA was 14 g/kWh. The results of a pilot study revealed that for the particular type of test engine with gradually retarded degrees of SOI (start of injection), MAO100 is in many cases comparable to most of the critical engine parameters and indicators, compared with D100 [21]. For some of the parameters (air mass flow, pressure provided by the turbocharger, air-fuel equivalence ratio, the level of nitrogen oxides, and slightly reduced oxygen content in the exhaust gases), depending on the load applied, the tests indicated a very small difference in measured values, that fell within the measurement error. In the form of increased levels of emission, negative trends were observed only for HC and $CO_2$ [21]. In general, the decreasing trends of soot concentration in the exhaust gases of microalgae oil were mostly associated with better carbon-oxygen balance, leading to an improvement in the combustion reaction and better promotion of soot oxidation, especially at the end of the cycle. The increasing trend of $CO_2$ gases for microalgae oil indicates the higher fuel consumption rates for MAO100 being predetermined by its lower heating value compared with diesel fuel. Another strong argument for a wider adoption of microalgae oil in diesel engines lies in the fact that when the test engine was powered by pure P.moriformis oil, the indicated thermal efficiency ($\eta_i$) was approximately equal to D100—this trend was inversely proportional to the BSFC character. The use of MAO100 led to the slightly higher, but falling within the margin of error, indicated thermal efficiency (0.355) in comparison to diesel fuel (0.350) at high load (0.8 MPa), followed by the similarly close values of $\eta_i$ at 0.6 MPa (0.350 vs. 0.345) and 0.4 MPa (0.325 vs. 0.320) loads [35]. This allows a wider window for energy and emission enhancement for industrial diesel engines using microalgae oil as fuel.

### 3.2. Model Calibration Outcomes

1-D predictive engine model was developed and calibrated. Developing MM for better accuracy needs a lot of data related to component characteristics and computational power. Data requirements vary with fuel characteristics, engine architecture, and depending on how detailed the model being developed was, based on the requirements and information available [36]. According to the engine IMPULS simulation model and the acquired operating points of 1Z engine running with pure diesel fuel and pure microalgae oil, the engine input and output performance database was obtained, including 16 variables juxtaposed for three different loads (see Table 4). The calibration of MM for the 1Z engine running on pure diesel fuel is described in detail in Ref. [23]. Measured in-cylinder pressure

is used to calculate the rate of heat release (see Figure 4). Variables such as $G_{air}$, $P_k$, $T_k$, $\varphi_{inj}$, and others were used to run IMPULS. 1-D predictive engine model also calculated an in-cylinder pressure; however, since that was a tool that also needs calibration, only the measured in-cylinder pressure was used. Hence, a calibration of constants for various form factor ($m$) and conditional duration of combustion ($\varphi_z$) values was performed to match the measurement data as accurately as possible. As described in Ref. [37], an automated calibration script was used for a set of 1Z operation points together with a range for the model constants. The regiment evaluated the output from IMPULS and compared it with measurement datasets. The result of this routine is provided as a new dataset of constants for further iterations. The whole procedure was enabled to stop automatically when the regiment meets one of its stopping criteria. The main output of MM calibration is a derivation of the optimum constants with the smallest difference between the measured and predicted results [37]. According to Stenberg (2008) [37], a set of constants are applied to the validation points to see the performance of MM for all feasible regions of an optimization problem rather than those used for the training of the MM.

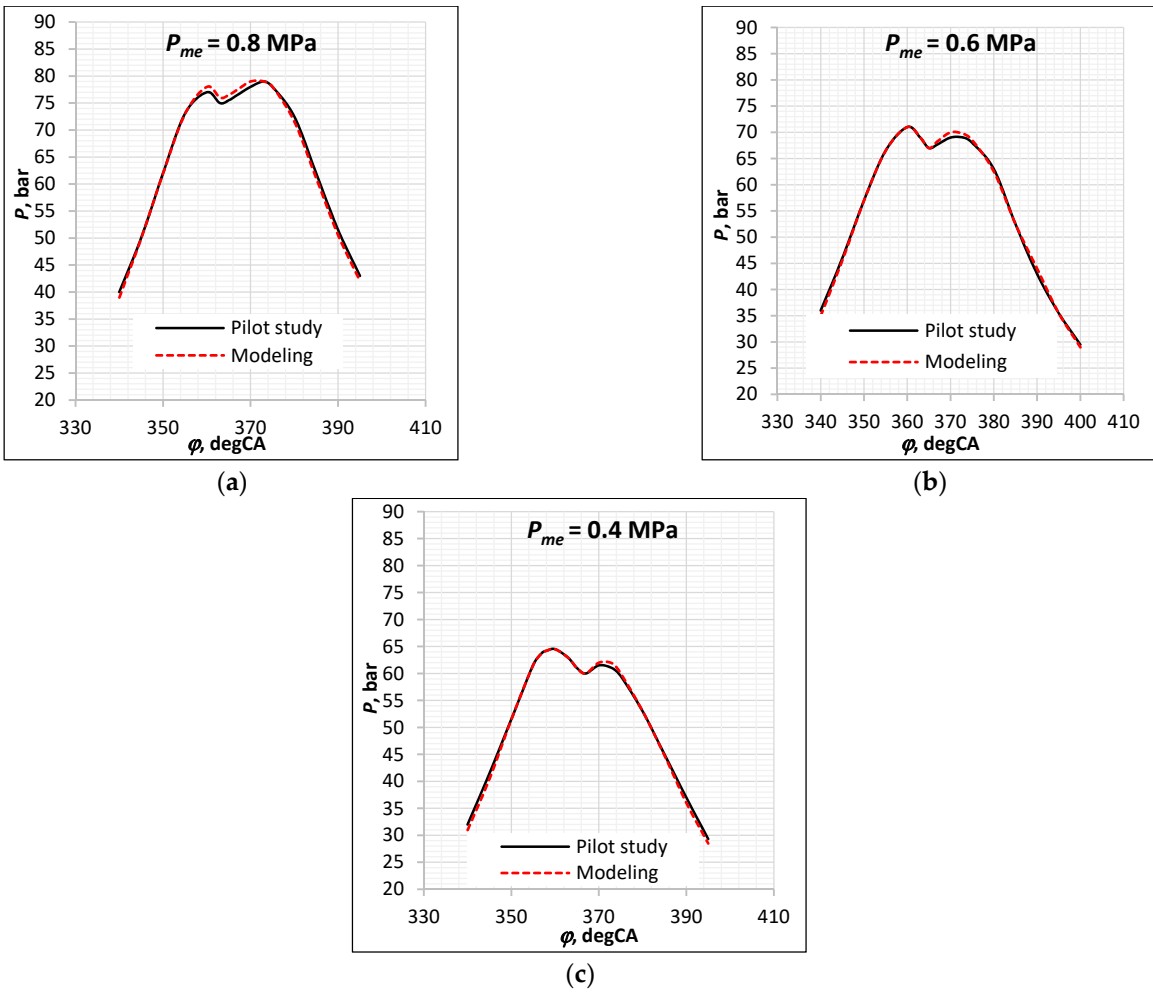

**Figure 4.** *Cont.*

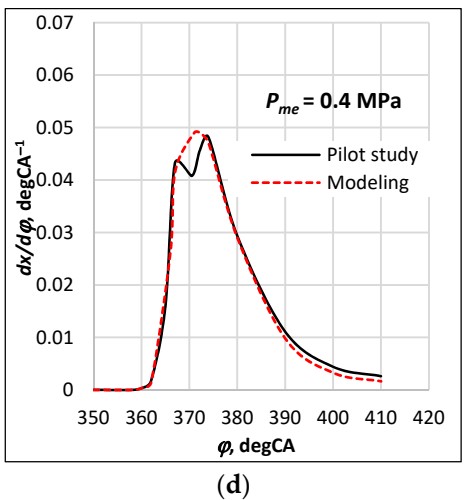
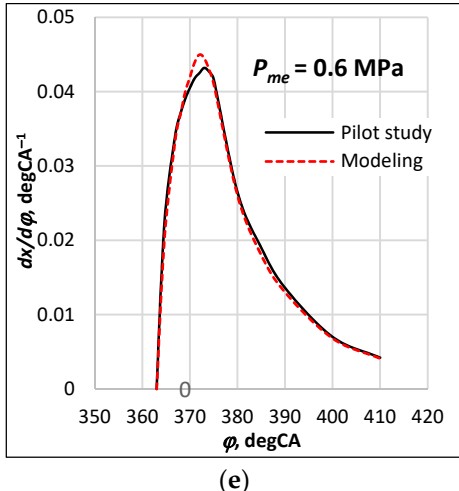

**Figure 4.** The combustion cycle of a diesel engine: juxtaposition of pilot study outcomes and mathematical modelling data ($\varphi_{inj}$ = 2 degCA BTDC, $n$ = 2000 min$^{-1}$): in-cylinder pressure (**a**–**c**) and heat release rate (**d**,**e**).

Whether the established model is accurate or not could be evaluated by the fitting degree between the modeling data and the experimental dataset. An obtained model accuracy for various parameters is as follows: $p_{me}$, 0–4.3%; $p_K$, 2.5–4.5%; $T_K$, 0.8–1.7%; $p_c$, 0.6–1.7%; $p_{max}$, 1.6–3.9%; $T_g$, 1.9–3.2%; $\alpha$, 5.1–10.0%; $\lambda$, 0–3.9%; $\eta_e$, 0.3–3.4%; and $g_{cycl}$, 0–1.7%. In general, the biggest difference between the measured and predicted parameters did not exceed 3.5% on average. The modelling results of the same parameters for average and high loads have shown a close to 100% percent coincidence with the experimentally obtained ones.

As described earlier, we simultaneously collected in-cylinder pressure data for three different loads coupled with the datasets of pressure provided by the turbocharger. Firstly, the variation in in-cylinder pressure was determined experimentally (see Figure 2) when the engine was running with both unary fuels, averaged over 10.000–13.300 cycles per each load setting. Then, the MM was calibrated for the reference point (75% load) of an engine running on diesel fuel [23] and microalgae oil using also some data from the manufacturer to select the appropriate maps for the AC compressor AFN TH 614 and turbine CHRA 454082 of the turbocharger due to the lack of their data. They are rescaled from existing maps to adjust the corrected mass flow rate and the corrected speed with the given pressure ratio exported from the in-cylinder pressure diagram to fit the current 1-D model [38]. This interim undertaking is important in calibrating the outlet pressure of the compressor by fitting the computed pressure diagram with the experimental one along the compression process. It also helps to verify the experimentally established amount of air and thus the volumetric efficiency against the modelling outcomes. This process is repeated for the other two cases presented in Figure 4 (0.6 MPa (50% load) and 0.4 MPa (25% load) for better calibration of the maps of the turbocharger and for improving the accuracy of the computed results along the different cases. The initial conditions of atmospheric pressure and temperature are adapted for each of the three loads, according to the pilot study data available for better accuracy [38].

**Table 4.** Juxtaposition of the pilot study data and MM results ($n$ = 2000 min$^{-1}$, fuel type: microalgae oil).

| $P_e$ kW | $p_{me}$ bar | | $\varphi_{inj}$ CAD TDC | $p_K$ bar | | $T_K$ K | | $\varphi_i$ CAD | | $p_c$ bar | | $p_{max}$ bar | | $T_g$ K | | $G_{air}$ kg/h | $G_f$ kg/h | | $\alpha$ | | $\lambda$ | | $\eta_i$ | $\eta_e$ | | $g_{cycl}/\eta_m$ | |
|---|---|---|---|---|---|---|---|---|---|---|---|---|---|---|---|---|---|---|---|---|---|---|---|---|---|---|---|
| 12.57 | 0.4 | 0.383 | −2.5 | 1.33 | 1.27 | 353 | 359 | 9.0 | 7.5 | 64.5 | 64.0 | 61.0 | 62.0 | 664 | 643 | 137.8 | 3.78 | 3.78 | 2.89 | 2.60 | 0.94 | 0.97 | 0.46 | 0.325 | 0.314 | 0.01573 | 0.01600 |
| 18.85 | 0.6 | 0.6 | −3.0 | 1.48 | 1.42 | 362 | 359 | 7.5 | 6.9 | 70.5 | 71.7 | 69.0 | 76.3 | 740 | 733 | 147.6 | 5.41 | 5.41 | 2.16 | 2.05 | 0.98 | 1.00 | 0.545 | 0.340 | 0.343 | 0.02253 | 0.02253 |
| 25.13 | 0.8 | 0.8 | −3.0 | 1.59 | 1.55 | 359 | 359 | 6.0 | 6.5 | 77.5 | 78.0 | 79.0 | 80.0 | 792 | 807 | 156.5 | 6.90 | 6.90 | 1.79 | 1.75 | 1.03 | 1.03 | 0.44 | 0.356 | 0.357 | 0.02876 | 0.02876 |

As previously described in Ref. [38], to minimize the mean absolute percentage deviation, we used a derivative version of the goodness-of-fit method that describes the correlation between the in-cylinder pressure diagram of a 1Z engine and the corresponding data from the pilot study along the combustion process, by finding the optimal values of injection system parameters due to the lack of their availability. Regarding the validation of MM, Figure 4 displays the comparison of experiments and numerical results for three load points of a 1Z engine running on pure microalgae oil. At the high load (0.8 MPa) operating point where the magnitudes of the brake mean effective pressure are greater, the values obtained during the pilot study and MM are within 1.3% of each other, and at the lower load points where the magnitudes of the BMEP are much smaller, the values are within 2.1% of each other for $P_{me}$ = 0.6 MPa and 1.6% for $P_{me}$ = 0.4 MPa (see Figure 4). The plots indicate a very good agreement between the measured and predicted values for brake mean effective pressure. It is worth nothing that the range of calculation errors (1.3–2.1%) was mainly characteristic to these parts of the curves, which show the pressure peaks exceptionally. Again, this is due to the magnitude of the BMEP values. In terms of peak pressure, the error remains lower than +/−1.5 bar and can be considered as acceptable. Overall, this work shows a methodology able to establish some parameters for optimum engine operation in high-pressure direct injection combustion mode. This was achieved through the analysis of combustion chamber pressure (Figure 4a–c) and heat release rate (Figure 4d,e). The resulting plots of heat release have high precision, leading to a reliable estimation of the process, that is critical to a good modeling of the selected types of diesel engines. The insignificant differences were obtained only around the piston position corresponding to the top dead center, which in turn was predetermined by the application of the single-phase vibe combustion model for diesel engines. In other words, microalgae oil performance was accurately approximated by a single linear segment, which means a single-phase form. The differences between the modelling and experimental outcomes of the heat release characteristics on average did not exceeded 2%, with the exception of $dx/d\varphi$ peaks that show the 5–7% discrepancy in values. Figure 4 illustrates that the simulation data of the cylinder pressure and the heat release rate were in good agreement with the pilot study results; thus, the model was reasonable.

The progress of the engine differential experimental values HRR ($dQ/d\varphi$) when switching the engine from diesel fuel to microalgae oil is shown in Figure 5. Commonly in compression ignition engines, the four stages in the combustion process are as follows: ignition delay period, premixed combustion phase (all fuel has been injected, and the pressure increases rapidly), mixing controlled combustion phase (the fuel is burned and produces power), and late combustion phase (the pressure is going down) [35].

In this section, the impact of microalgae oil and diesel fuel on engine combustion characteristics is discussed in terms of HRR. Depending on the research aims, the characterization of the ongoing processes within engine cylinders are calculated using the experimentally obtained in-cylinder process. These are the heat release characteristics $\frac{dx}{d\varphi} = f(\varphi)$, expressed by differential equations, and the parameters of the in-cylinder processes, such as ignition delay, factor cycle dynamics ($\sigma$, $\lambda$, $dp/d\varphi_{max}$), characteristic temperature and pressure ($T_{max}$, $P_{max}$), and others [39]. The $dx/d\varphi$ is considered as the basic parameter describing heat release during the process of fuel combustion within the cylinders. The reliability of the results of mathematical modelling of the diesel engine parameters depends on the appropriate setting of $dx/d\varphi$ [39]. HRR is affected by the amount of air-fuel ratio, SOI, and the thermodynamic properties inside the cylinder. The premixed phase of combustion ranges between the start of combustion (362 degCA for MAO100 and 363 degCA for D100) and the CA corresponding to the maximum value of heat flux (367 degCA for D100 and 374 degCA for MAO100) [21]. The early stage of the premixed combustion phase is associated with the rapid increase in in-cylinder pressure showing the highest peaks of HRR for diesel fuel at $P_{me}$ = 0.4 MPa: 38 J/degCA and 0.063 degCA$^{-1}$, (see Figure 5a,b). For the same load, MAO100 exhibited a longer duration of heat release during the premixed phase of combustion (~362.0 . . . 373.8 degCA) with the maximum value of

heat flux totaled at 29.5 J/degCA and 0.048 degCA$^{-1}$. Engine performance at average load (0.6 MPa) demonstrated much more even HRR trends in terms of exhibited character of the maximum heat flux (35.7 and 35.2 J/degCA as well as 0.042 and 0.043 degCA$^{-1}$ for D100 and MAO, respectively) if compared with $P_{me}$ = 0.4 MPa (Figure 5c,d). During the assessment of heat release characteristics for the selected range of loads (Figure 5), the $m$ and $\varphi_z$ parameters demonstrated, on average, 10% higher values in the case of microalgae oil if compared with D100 as the reference fuel. The explanation for this lies in a fact that, during the early stage of mixing controlled combustion phase, a smaller portion of MAO was prepared for combustion during the ignition delay period due to the lower LHV, which in turn reduced the HRR. The values of $m$ and $\varphi_z$ will be applied for the modelling of CAT 3512B HB-SC engine as described in Section 3.3. The combustion cycle of a diesel engine: juxtaposition of pilot study outcomes vs mathematical modelling ($\varphi_{inj}$ = 2 degCA BTDC, $n$ = 2000 min$^{-1}$).

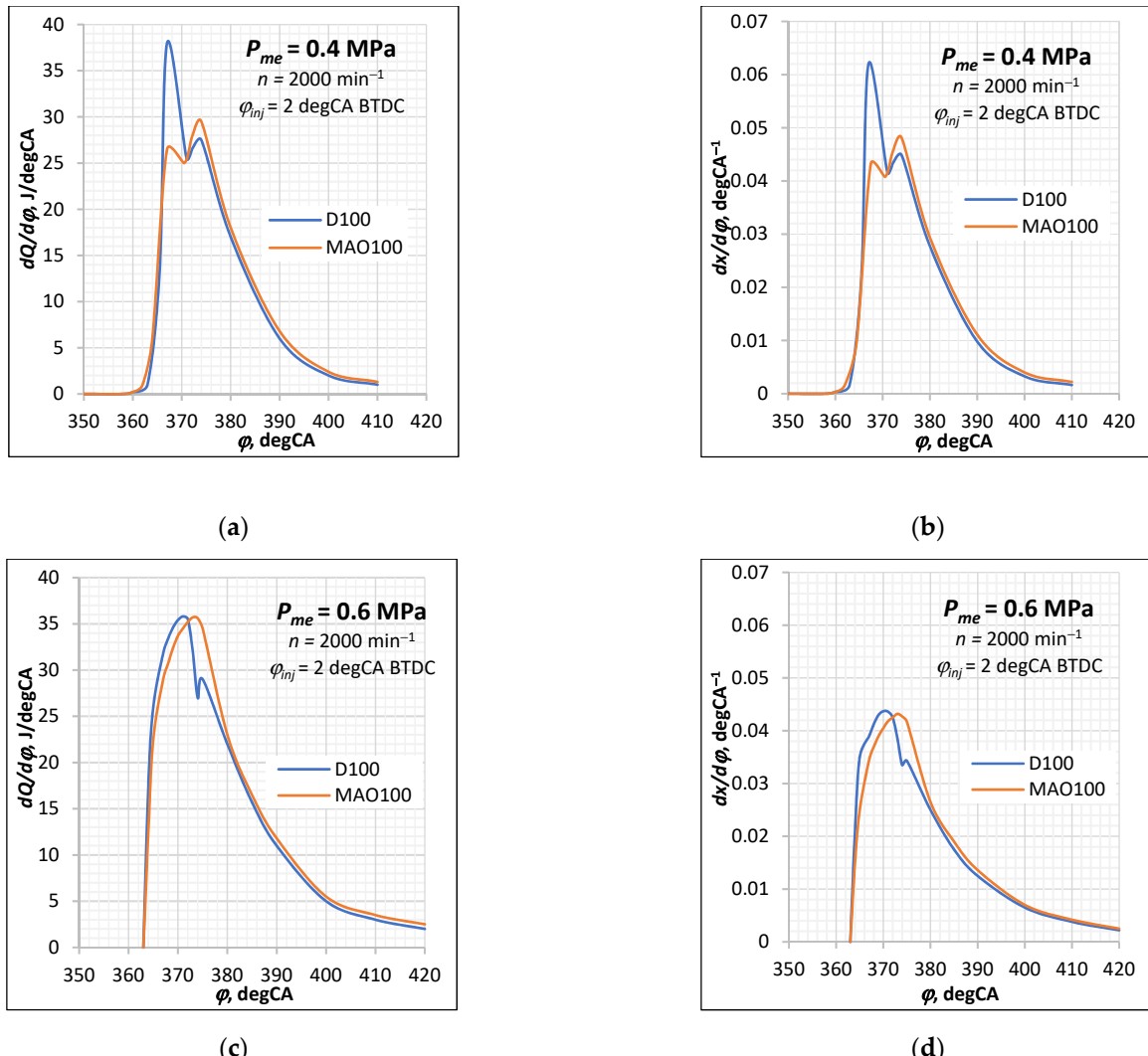

**Figure 5.** Changes in heat release characteristics of an 1Z engine when comparing diesel fuel and microalgae oil: at 0.4 MPa (**a**,**b**) and 0.6 MPa (**c**,**d**).

### 3.3. The Use of a Calibrated MM for the Assessment of CAT 3512B HB-SC Industrial Engine Running on Microalgae Oil

One of the most efficient methods to examine the critical challenges in the internal combustion engine research, when diesel engines of various types and modifications are subject of successful adoption of alternative and sustainable fuels, is the application of

modeling software, including IMPULS, in which important indicators of the HRR ($m$ and $\varphi_z$) were determined according to logarithmic anamorphosis method [40,41] and modified methodology by Bulaty and Glanzman [42]. Meanwhile, the successfully established heat release characteristics for the 1Z engine of a passenger car were successfully adopted to the industrial diesel engine CAT 3512B HB-SC, equipped with a similar fuel injection system. A list of variables experimentally obtained for the 1Z load close to the nominal regime, served as a prerequisite for MM to assess the brake effective mean pressure, indicated thermal efficiency, effective efficiency, brake-specific fuel consumption, exhaust gas temperature, excess air coefficient, maximum cycle pressure (combustion pressure), pressure after compression, mechanical efficiency, and the change in $CO_2$ emissions at six different speeds ($n$ = 1000, 1200, 1300, 1400, 1500, 1600, and 1800 min$^{-1}$) of a CAT 3512B HB-SC engine (see Table 5). For the simulation of CAT 3512B HB-SC by using the statistical data of an internal combustion engine, the datasets of $p_{max}$ (maximum combustion pressure) were assumed according to the nominal regime of an engine ($n$ = 1800 min$^{-1}$) running with diesel fuel as well as the threshold value of the exhaust gas temperature (before entering the turbine) was set to $T_g \leq 973$ K (700 °C) and served as a limiting parameter. In other words, MM calibration for all obtained parameters was performed at the nominal regime of a CAT engine, while, for the other seven speeds, the calibrating parameters were not further adjusted. Instead of this, the obtained parameters were juxtaposed with the calculated threshold values of $P_{max}$ and $T_g$ to obtain a clear view when it is necessary to limit the volume of $g_{cikl}$ as a way to prevent a further increase in process parameters if their threshold value was already achieved. The following indicative parameters were obtained for diesel fuel (I) and microalgae oil (II) at two different injection settings of an engine (Table 5). For engine operation with diesel fuel, we employed $m$ and $\varphi$ parameters that were established through the use of the modified methodology by Bulaty and Glanzman [42], while, for MAO100, these parameters were assumed to be 10% higher compared with D100. To prevent the rise of the exhaust gas temperature of an engine running with microalgae oil above the 973 K level (see Table 5), the third (III) mode of injection was a subject of IMPULS modeling, where the cyclic portion ($g_{cycl}$) of microalgae oil was reduced while other parameters ($m$ and $\varphi_z$) remained unchanged. For the in-cylinder pressure profiles ($P_{me}$) obtained at each selected speed, we apply the torque reserve factor of 1.1—a value that is typical to industrial diesel engines. A precondition of $g_{cycl} \times$ LHV = idem$_{D100}$ has been met, which is based on the already discussed above impermissible level of the exhaust gas temperature ($T_g$) caused by the prolonged fuel burning process and, especially, its higher intensity during the second phase of combustion at $n$ = 1000–1300 min$^{-1}$ (see Table 5).

**Table 5.** Indicative parameters of a CAT 3512B HB-SC engine: injection settings I, II, and III (the numbers highlighted in a red color indicate the $Tg$ values exceeding the 973 K level).

| | 1800 m$^{-1}$ I/II/III * | 1600 m$^{-1}$ I/II/III * | 1500 m$^{-1}$ I/II/III * | 1400 m$^{-1}$ I/II/III * | 1300 m$^{-1}$ I/II/III * | 1200 m$^{-1}$ I/II/III * | 1000 m$^{-1}$ I/II/III * |
|---|---|---|---|---|---|---|---|
| $p_{me}$, bar | 18.6/17.8/17.8 | 19.7/18.9/18.9 | 19.9/19.2/19.2 | 20.1/19.4/19.4 | 20.2/19.4/17.1 | 17.1/16.6/15.2 | 12.7/12.7/11.8 |
| $\eta_i$ | 0.451/0.432/0.432 | 0.458/0.442/0.442 | 0.456/0.440/0.440 | 0.452/0.436/0.436 | 0.447/0.432/0.438 | 0.447/0.435/0.439 | 0.445/0.437/0.440 |
| $\eta_e$ | 0.388/0.372/0.372 | 0.410/0.390/0.390 | 0.415/0.400/0.400 | 0.419/0.404/0.404 | 0.420/0.405/0.406 | 0.419/0.407/0.408 | 0.409/0.401/0.401 |
| $b_e$, g/kW/h | 217/262/262 | 205/248/248 | 203/245/245 | 201/241/241 | 200/241/240 | 201/240/240 | 206/244/244 |
| $T_t$, K | 889/904/904 | 903/919/919 | 920/935/935 | 944/963/963 | 969/996/972 | 970/991/970 | 972/994/971 |
| $\alpha$ | 2.07/2.03/2.03 | 1.89/1.87/1.87 | 1.80/1.79/1.79 | 1.69/1.68/1.68 | 1.59/1.57/1.59 | 1.49/1.47/1.51 | 1.41/1.39/1.43 |
| $p_{max}$, bar | 146/145/145 | 146/137/137 | 140/131/131 | 134/123.5/123.5 | 128/116/108 | 111/101/96.5 | 88.5/81/79 |
| $P_K$, bar | 3.70/3.70/3.70 | 3.24/3.28/3.28 | 3.05/3.09/3.09 | 2.84/2.87/2.87 | 2.66/2.65/2.33 | 2.08/2.10/1.95 | 1.47/1.47/1.43 |
| $\eta_m$ | 0.86/0.86/0.86 | 0.89/0.89/0.89 | 0.91/0.91/0.91 | 0.93/0.93/0.93 | 0.94/0.94/0.93 | 0.935/0.935/0.928 | 0.92/0.92/0.91 |
| $\delta CO_2$ | +4.0% | +4.5% | +4.8% | +4.2% | −8.5% | −4.5% | +1.0% |

* The third mode of injection with the reduced cyclic portion of microalgae oil.

The juxtaposition of (I) and (III) outcomes shows that, notwithstanding the fact that $T_g$ reached the desired level at $n$ = 1000–1300 min$^{-1}$ in the case of MAO100 (see Table 5), a 4.0–4.5% reduction in engine torque/brake mean effective pressure ($p_{me}$) was observed in the range of crankshaft speeds starting from its nominal value which corresponds to the OEM settings up to the speed that generates the maximum torque, followed by 7.1–15.3% decrease starting from $n$ = 1300 min$^{-1}$ ($M_{t\ max}$) and further on until $n$ = 1000 min$^{-1}$ (see Table 5). At the same time, brake-specific fuel consumption of MAO100 increased on average by 19–21% compared with diesel fuel. This phenomenon was predetermined by the difference between LHV values for pure diesel fuel (42.8 MJ/kg) and MAO100 (36.8 MJ/kg) that comprises 13.8% and making it necessary to increase the volume of $g_{cikl}$ to compensate the lower energy content of microalgae oil. These numbers suggest that practically it is essential to ensure that 15.96% larger amounts of fuel are delivered to each engine cylinder when microalgae oil is burnt [35]. Another reason for the exceeded fuel consumption lies in the decline of the effective pressure ($\eta_e$) in the speed range of $n \le$ 1300 min$^{-1}$ (due to restriction measures for exhaust gas temperature). Nevertheless, the precondition for the induction period $\varphi_i$ = idem was applied for both D100 and MAO100, microalgae oil was responsible for a longer duration of the combustion process that led to the appearance of the reserve next limiting factor—$P_{max}$ (maximum cycle pressure)—showing a decline on average by 9.0–10.5 bar along the range of higher speeds (1800–1400 min$^{-1}$), and by 14.5–20.0 bar for the range of lower speeds (1300–1200 min$^{-1}$). Two influencing factors had a direct impact on the obtained results: (i) adjustment of a combustion process (the uniform duration of the induction phase $\varphi_i$ led to a longer conditional combustion duration $\varphi_z$ in the case of MAO100), and (ii) forced reduction in $g_{cikl}$ to prevent further increase in $T_g$ above the threshold value of 973 K. If compared with diesel fuel, $CO_2$ emissions of the engine running with microalgae oil (full life-cycle carbon dioxide emissions are absent from analysis) have shown an increasing trend (by 4.0–4.5%) for the higher speed range of an engine (1800–1400 min$^{-1}$) and, on the contrary, for the lower speed range of 1300–1200 min$^{-1}$, the 4.5–8.5% reduction was observed.

From the simulation outcomes of the CAT 3512B HB-SC engine running with MAO100, which clearly shows the untapped margin of maximum permissible values of the restrictive criterion $P_{max}$ (see Table 5), we found reasonable grounds for an improvement of the dynamic characteristics of ICE via further advancement of SOI as depicted in Figure 3 (Section 3.1). Hence, the MM outcome using the (IV) injection setting for CAT 3512B HB-SC engine running with pure microalgae oil ($T_g \le$ 973 K, $\varphi_{inj}$ = 2 degCA BTDC) was juxtaposed against (I) diesel fuel and presented in Table 6. Figure 6 presents the variation in break mean effective pressure, maximum cycle pressure (combustion pressure), indicated thermal efficiency, and exhaust gas temperature for the entire range of engine speeds (1800–1000 min$^{-1}$) for the I (D100), II (MAO100), and IV (MAO100) injection strategies. Results revealed that the traction characteristics of a CAT 3512B HB-SC engine were significantly improved after the last correction of the injection settings. In comparing diesel fuel (I) and microalgae oil (IV), we found that the difference in indicated thermal efficiency now comprises only 0.7–2.0% (see Table 6) and the threshold value of 973 K for exhaust gas temperature was successfully reduced below the threshold level or, in the case of $n$ = 1300 min$^{-1}$, an obtained temperature level felt within the margin of experiment error (see Table 6 and Figure 6).

The mathematical modeling results of $P_{me}$, $P_{max}$, $\eta_i$, and $T_{ex}$ for the universal multipurpose diesel engine CAT 3512B HB-SC, which are reported in Figure 6, generally confirm the benefit of using the IV-th injection setting for pure microalgae oil ($T_g \le$ 973 K, $\varphi_{inj}$ = 2 degCA BTDC) over the II-nd one. A careful look at variation in $P_{me}$ and $P_{max}$ under the range of higher speeds (1800–1400 min$^{-1}$) have shown the increase in these parameters for the case IV on average by 1.6–2.2% and 1.4–6.7%, respectively (Figure 6a,c). The range of lower speeds (1300–1200 min$^{-1}$) was responsible for in 0.5–1.8% lower $P_{me}$ rates and in 6.0–10.8% higher $P_{max}$ values. It was found that at the regimes of external speed characteristics, the difference in $P_{max}$ and $P_{me}$ for MAO100 (IV) and D100 (I) did

not exceed 0.7–2.0% and 2–4%, respectively. The exhaust gas temperature was reduced by 8–12 K within the range of engine speeds starting from $n = 1400$ m$^{-1}$ to $n = 1800$ m$^{-1}$. A drastic reduction in $T_g$ by 18–40 K was observed for the IV-th setting of injection at $n = 1300$–1000 m$^{-1}$ (Figure 6d), which in turn enabled us to suppress the further rise of temperature above the threshold value of $T_g \leq 973$ K and to diminish its negative effects. Figure 6d shows the variation in indicated thermal efficiency against the entire range of engine speeds for the I, II, and IV injection strategies. If comparing the modelling outcomes for the II and IV injection strategies, the later demonstrated 0.4–2.3% higher $\eta_i$ values, thus being an indicator of better fuel conversion efficiency achieved through the advancement of SOI.

The modelling outcomes (Figure 6, Table 6) revealed that the IV-th injection strategy applied for microalgae oil is more or less equally sensitive to certain engine parameters ($P_{me}$, $P_{max}$, $\eta_i$, $\eta_e$, $\alpha$, $p_k$, $\eta_m$, and $T_{ex}$) compared with diesel fuel (I) as well as prevents temperature rise and BMEP losses—negative trends emerged from the modelling of the II-nd and III-rd scenarios. This makes a serious prerequisite for a more rigorous evaluation of the suitability of *P. moriformis* microalgae oil as fuel for FPT Industrial Cursor 13 engine family, thus bringing a potentially carbon-free resource into the mix, in the form of renewable fuel for the broader range of CI engines.

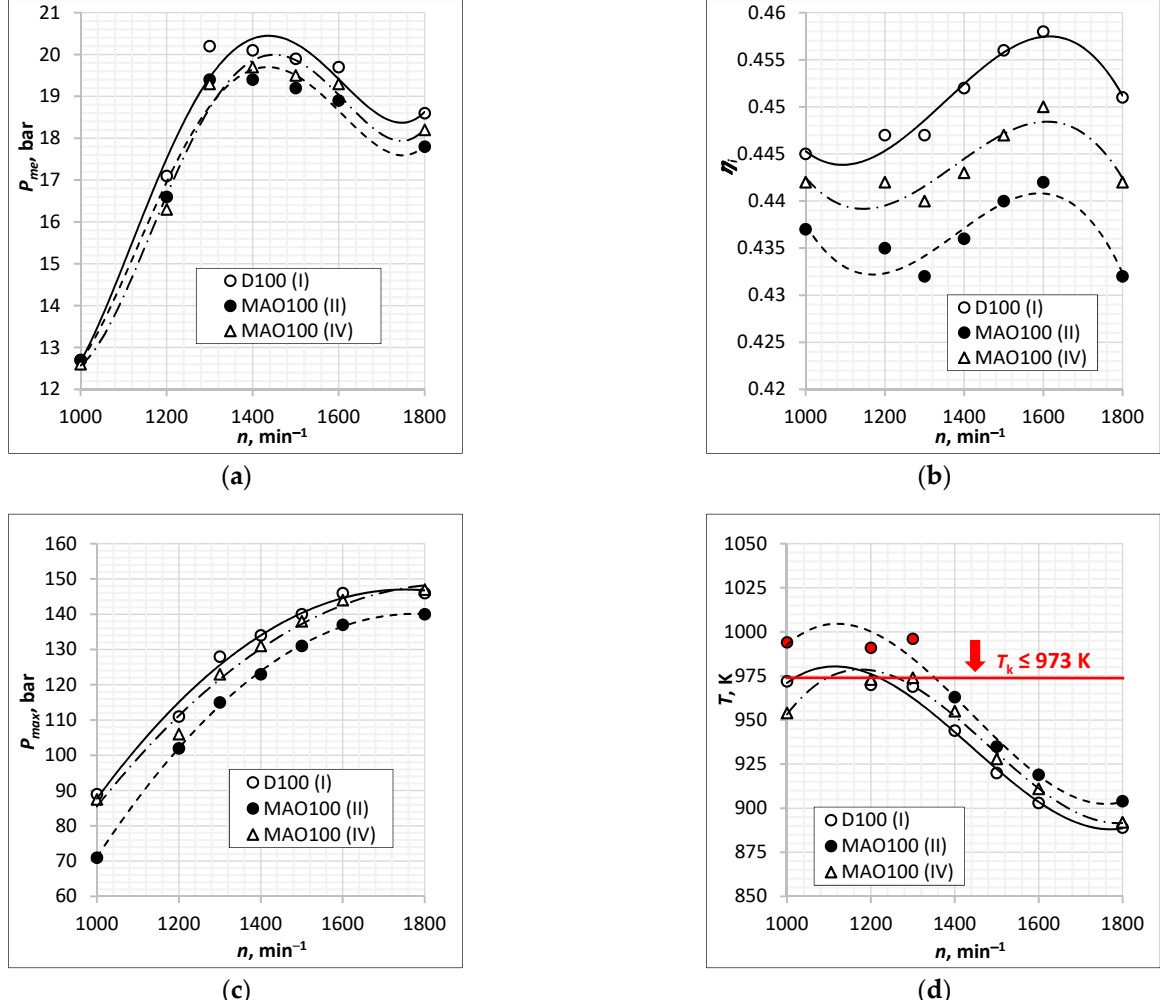

**Figure 6.** Variation in break mean effective pressure (**a**), indicated thermal efficiency (**b**), maximum cycle pressure (combustion pressure) (**c**), and exhaust gas temperature (**d**) for the entire range of engine speeds (1800–1000 min$^{-1}$) for the I, II, and IV injection strategies.

**Table 6.** Indicative parameters of a CAT 3512B HB-SC engine: injection settings I and IV (the numbers highlighted in a red color indicate the obtained improvement in $T_g$ values within the depicted range of speed).

| | 1800 m$^{-1}$ D100/MAO100 | 1600 m$^{-1}$ D100/MAO100 | 1500 m$^{-1}$ D100/MAO100 | 1400 m$^{-1}$ D100/MAO100 | 1300 m$^{-1}$ D100/MAO100 | 1200 m$^{-1}$ D100/MAO100 | 1000 m$^{-1}$ D100/MAO100 |
|---|---|---|---|---|---|---|---|
| $p_{me}$, bar | 18.6/18.2 | 19.7/19.3 | 19.9/19.5 | 20.1/19.7 | 20.2/19.3 | 17.1/16.3 | 12.7/12.6 |
| $\eta_i$ | 0.451/0.442 | 0.458/0.450 | 0.456/0.447 | 0.452/0.443 | 0.447/0.440 | 0.447/0.442 | 0.445/0.442 |
| $\eta_e$ | 0.388/0.380 | 0.410/0.402 | 0.415/0.407 | 0.419/0.411 | 0.420/0.413 | 0.419/0.413 | 0.409/0.406 |
| $b_e$, g/kW/h | 217/257 | 205/244 | 203/240 | 201/238 | 200/237 | 201/237 | 206/240 |
| $T_t$, K | 889/892 | 903/911 | 920/928 | 944/955 | 969/974 | 970/973 | 972/954 |
| $\alpha$ | 2.07/2.03 | 1.89/1.87 | 1.80/1.78 | 1.69/1.67 | 1.59/1.57 | 1.49/1.47 | 1.41/1.40 |
| $p_{max}$, bar | 146/147 | 146/144 | 140/138 | 134/131 | 128/123 | 111/106 | 88.5/87.5 |
| $P_K$, bar | 3.70/3.70 | 3.24/3.26 | 3.05/3.07 | 2.84/2.86 | 2.66/2.61 | 2.08/2.03 | 1.47/1.49 |
| $\eta_m$ | 0.86/0.86 | 0.89/0.89 | 0.91/0.91 | 0.93/0.93 | 0.94/0.94 | 0.935/0.935 | 0.92/0.92 |

### 3.4. Prognostic Assessment of Industrial Diesel Engine Family for Energy Efficiency and CO$_2$ Levels

This work sought to further understand the engine efficiency and prospects for improvement in heavy-duty diesel engines having a comparatively high compression ratio (and brake mean effective pressure, $P_{me}$) and broad field of application—the necessary prerequisites for facilitating an uptake and ensuring wider deployment of CO$_2$-neutral fuel, pure microalgae oil *P. moriformis*. The research approach involved the modelling and analysis of a heavy-duty diesel engine, representing the entire FPT Industrial Cursor 13 engine family, to evaluate the compatibility of microalgae oil with a wide range of industrial CI engines. The FPT Industrial Cursor 13 family engines comply with statutory emission standards since they have an optimized combustion process with quick and complete combustion that allows high levels of efficiency even with a lean gas mixture. The performance of the motors ranges from 180 to 420 kW and these power units are broadly applied for mobile (tractors, forestry machinery) and stationary applications (gen-sets). Engines of this family can be operated on diesel fuel and on natural gas. For the tuning of parameters of the Wibe combustion model included in 1-D thermodynamic engine cycle simulation tool IMPULS, we selected a water-cooled six-cylinder direct injection diesel engine of 353 kW capacity at rated speed $n$ = 1900 min$^{-1}$. The two primary reference engines tested were a 1Z model 1.9-L passenger car diesel engine, a representative engine for Volkswagen Group, and a CAT 3512B HD-SC model 51.8-L heavy-duty engine, representative of Caterpillar. A simulation study of engine performance taking into account different ratios of heat release parameters $m$ and $\varphi_z$ was performed ($m$ = 0–1.5, $\varphi_z$ = 50–80 degCA). To determine the energy efficiency indicator $\eta_i$ of a CAT 3512B HD-SC running on diesel fuel and microalgae oil ($p_{me}$ = idem) in a given load mode, it is sufficient to evaluate the air excess coefficient ($\alpha$) that has a direct influence on the duration of $\varphi_z$. Air-access coefficient values ($\alpha$ = 1.8–2.2) were taken from a range, which corresponds to the values provided in a technical engine specification. Model calibration and validation during the initial stages of this study enabled us to reach the difference threshold of 2–3% between the simulation outcomes of the main energy indicators ($\eta_i$, $\eta_e$) and the same parameters provided in the technical documentation from OEM. The outcomes of the prognostic assessment, in a form of relative change in the main FPT Industrial Cursor 13 engine family parameters (changes ($\delta$) of: indicated thermal efficiency ($\eta_i$), hourly fuel consumption ($G_f$), maximum cycle pressure ($P_{max}$), exhaust gas temperature ($T_g$), and CO$_2$ emissions) were presented in Table 7 and Figure 7. Table 7 contains the calculation outcomes of $\eta_i$, $G_f$, $P_{max}$, and $T_g$ for different $\varphi_z$ and $m$ combinations (48 combinations, 16 points of each parameter per single chart) as well as the relative changes of these parameters when diesel fuel and microalgae oil are compared. The dataset obtained for an engine's indication of the thermal efficiency when diesel fuel was used (see Table 7) served as a mean to build a reference chart (Figure 7a) to

compare and to contrast the changes of other parameters when the engine was switched to run with MAO100 (Figure 7b–f).

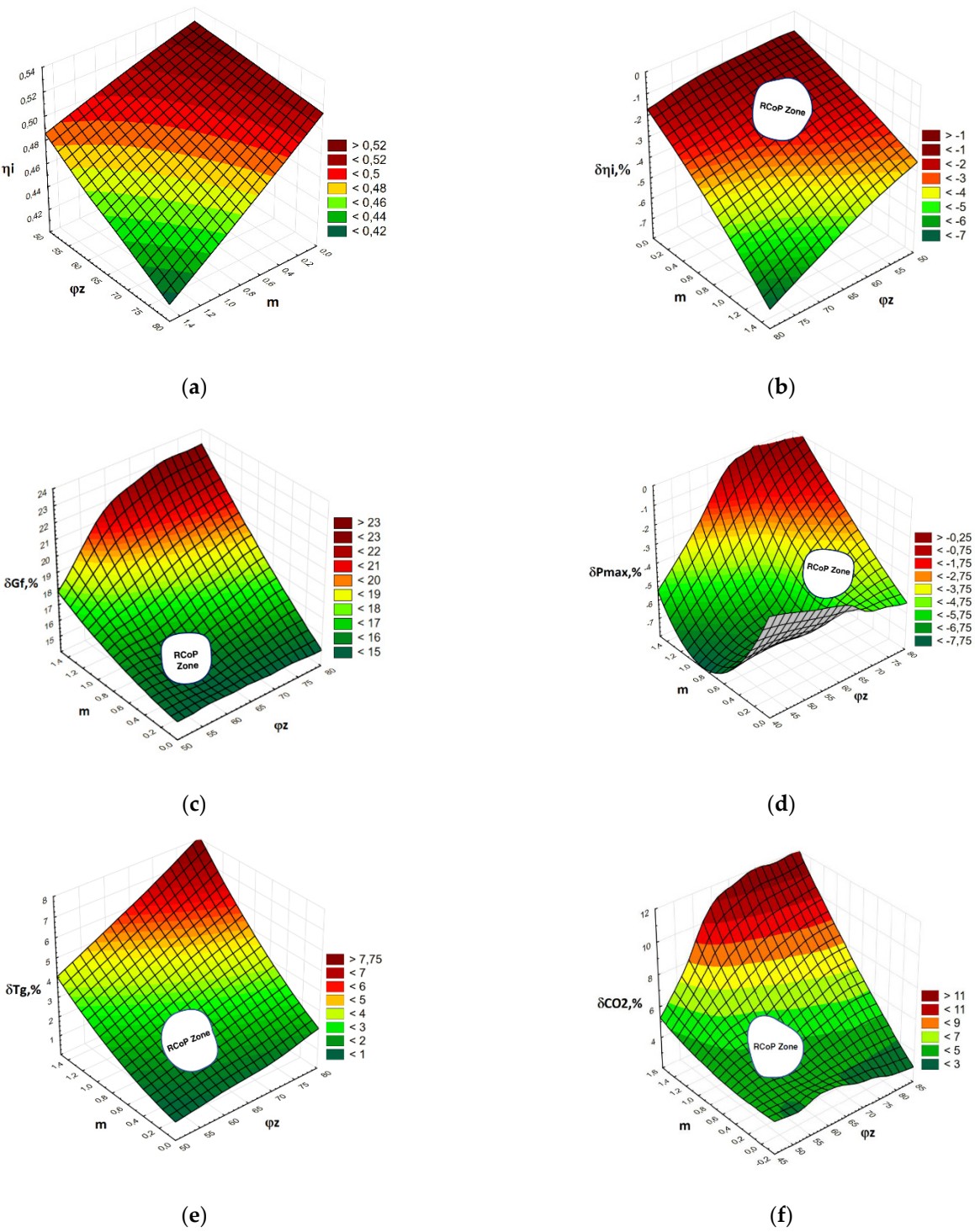

**Figure 7.** A relative change in engine operational parameters when switching from diesel fuel to microalgae oil: (**a**) Indicated efficiency of an engine fueled by diesel fuel; (**b**) Indicated efficiency (change); (**c**) Hourly fuel consumption; (**d**) Maximum cycle pressure (combustion pressure); (**e**) Exhaust gas temperature; (**f**) $CO_2$ emissions; RCoP Zone—the zone of rational combination of *m* and $\varphi_z$ parameters).

**Table 7.** Variation in engine parameters with different $m$ and $\varphi_z$ combinations (↑ indicates an increase in value).

| | $\eta_i/g_{cikl}$ | | | | | | | | $P_{max}/T_g$ | | | | | | | |
|---|---|---|---|---|---|---|---|---|---|---|---|---|---|---|---|---|
| $\varphi_z$ | $m$ | $m+10\%$ | $m$ | $m+10\%$ | $m$ | $m+10\%$ | $m$ | $m+10\%$ | $m$ | $m+10\%$ | $m$ | $m+10\%$ | $m$ | $m+10\%$ | $m$ | $m+10\%$ |
| | 0 | 0.05 | 0.5 | 0.55 | 1.0 | 1.1 | 1.45 | 1.6 | 0 | 0.05 | 0.5 | 0.55 | 1.0 | 1.1 | 1.45 | 1.6 |
| Diesel fuel ($\varphi_{inj}$ = 0 degCA TDC) | | | | | | | | | | | | | | | | |
| 50 | 0.5275 | 0.195 | 0.516 | 0.200 | 0.500 | 0.207 | 0.485 | 0.213 | 220 | 927 | 182 | 970 | 153 | 1018 | 133 | 1058 |
| 60 | 0.523 | 0.197 | 0.505 | 0.204 | 0.480 | 0.215 | 0.460 | 0.218 | 212 | 940 | 170 | 999 | 140 | 1067 | 129 | 1121 |
| 70 | 0.518 | 0.199 | 0.492 | 0.209 | 0.464 | 0.222 | 0.438 | 0.235 | 205 | 956 | 161 | 1031 | 132 | 1116 | 129 | 1193 |
| 80 | 0.513 | 0.201 | 0.481 | 0.214 | 0.445 | 0.231 | 0.415 | 0.248 | 199 | 971 | 154 | 1065 | 129 | 1171 | 129 | 1270 |
| Microalgae oil ($\varphi_{inj}$ = 0 degCA TDC, $\varphi_z$ is in 10% ↑ compared to D100) | | | | | | | | | | | | | | | | |
| 55 | 0.523 | 0.224 | 0.509 | 0.232 | 0.486 | 0.241 | 0.469 | 0.251 | 212 | 937 | 172 | 984 | 141 | 1047 | 129 | 1101 |
| 66 | 0.519 | 0.226 | 0.495 | 0.237 | 0.466 | 0.252 | 0.440 | 0.266 | 203 | 952 | 161 | 1027 | 130 | 1106 | 129 | 1180 |
| 77 | 0.512 | 0.229 | 0.481 | 0.244 | 0.444 | 0.264 | 0.413 | 0.284 | 195 | 970 | 153 | 1061 | 129 | 1170 | 129 | 1272 |
| 88 | 0.504 | 0.291 | 0.467 | 0.251 | 0.422 | 0.278 | 0.386 | 0.305 | 189 | 992 | 146 | 1101 | 129 | 1241 | 129 | 1372 |
| Juxtaposition of D100 ($\varphi_{inj}$ = 0 degCA TDC) vs. MAO100 ($\varphi_{inj}$ = 0 degCA TDC, $\varphi_z$ is in 10% ↑ compared to D100) | | | | | | | | | | | | | | | | |
| 50 | −1 | +15 | −1.5 | +16 | −2.8 | +16.4 | −3.5 | +17.8 | −3.7 | +1 | −5.5 | +1.4 | −7.8 | +2.8 | −3 | +4 |
| 60 | −1 | +15 | −2 | +16 | −3 | +17.2 | −4.5 | +22 | −4.3 | +1.3 | −5.3 | +2.2 | −7.3 | +3.7 | 0 | +5.3 |
| 70 | −1.2 | +15 | −2.5 | +16.7 | −4.3 | +19 | −5.7 | +23 | −5 | +1.5 | −5 | +2.9 | −2.3 | +4.8 | 0 | +6.6 |
| 80 | −1.8 | +15 | −3 | +17.3 | −5.2 | +20.3 | −7 | +23 | −5 | +2.2 | −5.2 | +3.3 | 0 | +6 | 0 | +8 |

The obtained results revealed that, if considering the smallest changes in the indicated thermal efficiency values as an outcome of the best compatibility of $m$ and $\varphi_z$ parameters, this indicator can be characterized by a relatively short period of heat release (50–60 degCA) and moderate dynamics ($m$ = 0–0.5).

Combination of these parameters in similar proportions is characteristic to industrial heavy duty diesel engines equipped with an accumulator fuel-injection system common rail [23]. If comparing the changes in indicated efficiency ($\delta\eta_i$) for the reference fuel and alternative fuel, the differences obtained at $m$ = 0.5 did not exceed 2–3% (Figure 7b).

Further increasing the time of conditional combustion duration ($\varphi_z$) up to 70–80 degCA, a range that is characteristic to many sorts of conventional injection systems in industrial CI engines, also gives a qualitatively positive non-changing impact of $m$ factor on $\eta_i$. However, quantitatively, the change in the indicated efficiency values reaches 6-7% (Figure 7b); this forewarns about the necessity of advancing the engine's timing ($\varphi_{inj}$) in order for the injection process to occur earlier. A curved up and elevated area of the plot indicates the 4–6% change in $P_{max}$ values (Figure 7d) if compared with diesel fuel, and the expected drop in exhaust gas temperature $T_g$ (Figure 7e) predetermines the increase in energy efficiency indicators without compromising the cylinder-piston group parts and turbine parts, that might be affected by thermal and mechanical overloads.

The differences in $CO_2$ emissions (full life-cycle carbon dioxide emissions are absent from analysis) within the zone of rational combination of $m$ and $\varphi_z$ parameters did not exceed 4–5% (Figure 7f) if juxtaposed to diesel fuel. Furthermore, an increase in $\varphi_z$ up to 70–80 degCA, that indeed led to the deterioration of the indicated efficiency of the engine, resulted in higher $CO_2$ emission levels by about 5.5–6.0%.

In general, the nature of the combustion process of conventional and alternative fuels is determined by the quality of the fuel spray and its distribution and mixing within the combustion chamber [43,44]. Breakup and distribution of the spray are largely determined by the physical and chemical properties of the fuel, air motion, and temperature that occur within the cylinders, coupled with injection pressure, nozzle design and geometry, and spray angle [42]. Heavy duty, industrial and passenger car diesel engines still require fuel injection equipment to be adopted according ever-renewing stringent requirements, especially for efficient operation at lower load and speed operating points. Since emission legislation drive cycles and EU and US federal exhaust emission standards for stationary engines require operation within these regimes, research into the qualitative variation in the working process parameters within diesel engines of various types and modifications is of great importance. At the same time, the possibility to achieve a desirable characteristic of fuel combustion for the selected compression ratio of an engine is limited by the magnitude

of the thermal and mechanical stress acting on the internal parts of ICE. This conclusion is based on the detailed monitoring of the obtained $P_{me}$ and $T_g$ values describing the different cases of the heat release process. For the selected value of the indicated efficiency $\eta_i = 0.48 - 0.49$ of a Cursor 13 engine, two different combinations of $\varphi_z$ and $m$ parameters derived from a function $X = f(\varphi)$ ($\varphi_z = 60$–70 degCA, $m = 0.5$ and $\varphi_z = 60$ degCA, $m = 1$) may be practically realized to achieve the desirable level of maximum combustion pressure $P_{max} = 130$–150 bar (at $\alpha \sim 2.0$). When switching from diesel to MAO100, it is expected that the value of the indicated engine efficiency will drop by 2–3%, however, an existing reserve in $P_{max}$ that comprises 5–7% (7–10 bar for the particular level of $P_{max} = 130$–150 bar) will open up room for further optimization of injection timing. These examples were embedded in Figure 7b–f, where the RCoP zones were identified for each operational parameter of the engine to facilitate the smooth transition to microalgae oil. Realization of the interim steps described in this study lays down the foundation for the accurate prognostic assessment of the expected operational parameters and $CO_2$ emissions for the engine family of similar design and a comparable range of $P_e$ values for widespread adoption of MAO100 usage.

## 4. Conclusions

The authors of this study raise a concern regarding the mass usage of fossil fuels mined from ancient deposits and consumed by industrial engines which are extensively used in various industries and sectors of the economy by offering to replace conventional fuel as a constituting part of the final energy mix with a novel type of biofuel produced from the less investigated microalgae specie *P. moriformis*. The following conclusions can be drawn as a result of the research:

- The interval of $-2 \dots 0$ degCA was found to be the best setting of an engine for smoke and $NO_x$ stabilization and reduction, nevertheless D100 or MAO100 were used. That leaves many opportunities for the wider deployment of their binary blends of various ratios to be consumed in diesel engines. Moreover, the pilot study showed that the use of microalgae oil in passenger car engine positively affected the indicated thermal efficiency ($\eta_i$) of the prime mover, finding it very similar to that of diesel fuel: 0.355 and 0.350 ($P_{me} = 0.8$ MPa), 0.350 and 0.345 ($P_{me} = 0.6$ MPa), 0.325 and 0.320 ($P_{me} = 0.4$ MPa).
- Following accuracy of the 1-D predictive engine model was obtained for various parameters: $p_{me}$: 0–4.3%, $p_K$: 2.5–4.5%, $\alpha$: 5.1–10.0%, $\lambda$: 0–3.9%, $\eta_e$: 0.3–3.4% and $g_{cycl}$: 0–1.7%, $T_K$: 0.8–1.7%, $p_c$: 0.6–1.7%, $p_{max}$: 1.6–3.9%, $T_g$: 1.9–3.2%.
- For the CAT 3512B HB-SC engine running with microalgae oil, we proposed a boundary condition for the injection modelling settings ($T_g \leq 973$ K, $\varphi_{inj} = 2$ degCA BTDC) that led to improvement of the overall traction characteristics: the difference in $\eta_i$ was almost eliminated and comprised only 0.7–2.0% without any compromise in exceeding the threshold value of 973 K for exhaust gas temperature.
- An extensive simulation of the FPT family engine, type Cursor 13 was performed by taking into account different strategies of a combustion process duration and its dynamics through the adjustment of m and $\varphi_z$ parameters within the broad range of variation: m = 0–1.5, $\varphi_z = 50$–80 degCA. The obtained results revealed that, if considering the smallest changes in the indicated thermal efficiency values as an outcome of the best compatibility of m and $\varphi_z$ parameters, this indicator can be characterized by a relatively short period of heat release (50–60 degCA) and moderate dynamics (m = 0–0.5).
- The zones of rational combination of m and $\varphi_z$ were identified for each operational parameter of the engine to facilitate the smooth transition to microalgae oil. It was found that the differences in carbon dioxide emissions within the zone of rational combination of m and $\varphi_z$ parameters did not exceed 4–5% if compared with D100.
- The study found that microalgae oil is more or less equally sensitive to key engine parameters, compared with diesel fuel, and can be successfully adopted to the entire families of industrial diesel engines.

**Author Contributions:** Conceptualization, S.L. and L.R.; methodology, S.L. and L.R.; software, S.L.; validation, S.L.; formal analysis, S.L. and L.R.; investigation, S.L. and L.R.; resources, S.L. and L.R.; writing—original draft preparation, L.R.; writing—review and editing, S.L. and L.R.; visualization, L.R. All authors have read and agreed to the published version of the manuscript.

**Funding:** This research received no external funding.

**Conflicts of Interest:** The authors declare no conflict of interest.

## Nomenclature

**Latin symbols**

| | |
|---|---|
| $C$: | Function parameter that is equal to 6.9 for the case of complete combustion |
| $dQ_e$: | Energy exchange (wall heat transfer from the cylinder gas) (J) |
| $dQ_{re}$: | Combustion heat released (J) |
| $dU$: | Change in internal energy in the system (J) |
| $g_{cycl}$: | Cyclic portion |
| $G_f$: | Hourly fuel consumption (kg/h) |
| $m$: | Form factor |
| $m_{ex}$: | Mass of exhaust gas (kg) |
| $m_{inj}$: | Mass of injected fuel (kg) |
| $m_s$: | Supply (intake) air mass (kg) |
| $n$: | Engine speed (rpm) |
| $P_e$: | Brake power (kW) |
| $p_{max}$: | Maximum cycle pressure (combustion pressure) (bar) |
| $p_k$: | Air pressure after compression (bar) |
| $p_c$: | Pressure of compression in the cylinder (bar) |
| $p_{me}$: | Brake mean effective pressure (bar) |
| $p_{mi}$: | Indicated mean pressure (bar) |
| $pdV$ | Volumetric work |
| $Q$: | Total heat input |
| $R$: | Gas constant (J/kg·K) |
| $T$: | Temperature (K) |
| $T_{ex}$: | Exhaust gas temperature before turbine (K) |
| $T_g$: | Exhaust gas temperature (K) |
| $T_K$: | Air temperature after compression (K) |
| $T_{max}$: | Maximum combustion temperature (K) |

**Greek symbols**

| | |
|---|---|
| $\alpha$: | Air-access coefficient |
| $\frac{dQ}{d\varphi}$: | Heat release rate (kJ/deg) |
| $\eta_m$: | Mechanical efficiency |
| $\eta_i$: | Indicated thermal efficiency |
| $\eta_e$: | Effective efficiency |
| $\tau$: | Time (s) |
| $\tau_z$: | Relative time of combustion |
| $\delta$: | Relative change |
| $\lambda = \frac{p_{max}}{p_c}$: | In-cylinder pressure rise rate |
| $X = f(\varphi)$: | Relative heat release ratio (°CA) |
| $\varphi_{inj}$: | High reaction fuel injection time (°CA) |
| $\varphi_z$: | Conditional combustion duration (°CA) |

**Abbreviations**

| | |
|---|---|
| BMEP: | Brake effective mean pressure (bar) |
| BTDC: | Before top dead center |
| BTL: | Biomass-to-liquid |
| CA: | Crankshaft rotation angle |
| CNG: | Compressed natural gas |
| CO: | Carbon monoxide |
| $CO_2$: | Carbon dioxide |

| | |
|---|---|
| D100: | Pure diesel fuel |
| degCA: | Crankshaft rotation angle degrees |
| DI: | Direct injection |
| GTL: | Gas-to-liquid |
| HHV: | Higher heating value (MJ/kg) |
| HRR: | Heat of release rate (J/degCA) |
| LHV: | Lower heating value (MJ/kg) |
| LNG: | Liquefied natural gas |
| LPG: | Liquefied petroleum gas |
| MAO100: | Pure microalgae oil |
| MM: | Mathematical model |
| $NO_x$: | Nitrogen oxides |
| OEM: | Original equipment manufacturers |
| RCoP Zone: | Zone of rational combination of m and $\varphi_z$ parameters |
| SOI: | Start of injection |
| TDC: | Top dead center |

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
