# Peer review of "Prognostic Assessment of the Performance Parameters for the Industrial Diesel Engines Operated with Microalgae Oil"

_sustainability, doi:10.3390/su13116482_

Round 1

Reviewer 1 Report

The paper broadly dealing with system level performance of the engine and apart from the 1-D model, not so much physical interpretation is involved. The findings are important for the high level decisions regarding new fuels for CI engines. 

a few comments: a large number of abbreviations have been used that all readers may not be familiar with. so authors need to introduce them     sentence on line 84-to 87 needs reference. sentence ending on line 90 also needs a reference to the previous limited research. there might be better ways to show results in figure 3. especially the variation with CA may need to be represented and explained in another figure for more clarification. line 298: the authors need to explain unreasonable points and their origin. regarding figure 6, the authors need to explain the reasons behind the diverse behavior of differences between fuels.

Author Response

I would like to thank to Reviewer#1 for the careful and detail review. Answers and some additional explanations for the mentioned queries are given in Response to Reviewer file. 

The paper is now revised substantially according to the reviewers’ comments (highlighted in a PDF file). We took on board all comments and have included necessary information. Hence, all comments have been directly or indirectly addressed.

Reviewer 2 Report

Manuscript Number: sustainability-1223976

Title: Prognostic assessment of the performance parameters for the industrial diesel engines operated with microalgae oil

This manuscript assessed the use of microalgae oil on three engines. The manuscript is generally well written and organised. This reviewer recommend a minor revision based on the following comments:

  1. Abstract needs to be rewritten. Current abstract only talked about what had been done, but not what had been found.
  2. Abbreviations should be defined when they first appear in the text.
  3. Figure 4: please add heat release rate curves to further validate the model.
  4. The conclusions should summarise the key findings of this study in a concise manner. Therefore, it is suggested to use bullet points to briefly summarise the key findings. Current conclusions are too wordy/lengthy, and talked too much of what had been done.

To me, the manuscript is novel and well written. The main problem is that the manuscript is too lengthy, and thus may be hard for readers to quickly get the main contents of the manuscript.

Author Response

I would like to thank to Reviewer#2 for the careful and detail review. Answers and some additional explanations for the mentioned queries are given in Response to Reviewer file. 

The paper is now revised substantially according to the reviewers’ comments (highlighted in a PDF file). We took on board all comments and have included necessary information. Hence, all comments have been directly or indirectly addressed.

Round 2

Reviewer 1 Report

The authors have met my concerns